# Thermosensitive Nanosystems Associated with Hyperthermia for Cancer Treatment

**DOI:** 10.3390/ph12040171

**Published:** 2019-11-25

**Authors:** Isabela Pereira Gomes, Jaqueline Aparecida Duarte, Ana Luiza Chaves Maia, Domenico Rubello, Danyelle M. Townsend, André Luís Branco de Barros, Elaine Amaral Leite

**Affiliations:** 1Faculdade de Farmácia, Universidade Federal de Minas Gerais, 31279-901 Belo Horizonte, Brazil; 2Department of Nuclear Medicine, Radiology, Neuroradiology, Medical Physics, Clinical Laboratory, Microbiology, Pathology, Trasfusional Medicine, Santa Maria della Misericordia Hospital, 45100 Rovigo, Italy; 3Department of Drug Discovery and Pharmaceutical Sciences, Medical University of South Carolina, Charleston, SC 29425, USA

**Keywords:** hyperthermia, thermosensitive systems, cancer treatment

## Abstract

Conventional chemotherapy regimens have limitations due to serious adverse effects. Targeted drug delivery systems to reduce systemic toxicity are a powerful drug development platform. Encapsulation of antitumor drug(s) in thermosensitive nanocarriers is an emerging approach with a promise to improve uptake and increase therapeutic efficacy, as they can be activated by hyperthermia selectively at the tumor site. In this review, we focus on thermosensitive nanosystems associated with hyperthermia for the treatment of cancer, in preclinical and clinical use.

## 1. Introduction

Cancer is considered a public health problem due to the high incidence and mortality. The World Health Organization (WHO) estimates 27 million cases of cancer and 17 million deaths from this disease for the year 2030 [1]. Currently, conventional chemotherapy regimens have limitations, such as low specificity, which generates adverse effects that compromise the treatment and the health of the patient. Targeted drug delivery systems are emerging as a powerful strategy to overcome the toxicity that can limit the successful treatment of cancer patients. Specifically, thermosensitive nanomaterials are promising in the treatment of cancer because of their ability to act at specific sites when associated with hyperthermia [2,3,4]. Thermosensitive carrier systems are composed of lipids or polymers that transition from the gel phase to the crystalline liquid phase in response to heat, thus allowing drug release specifically in the heated region [5]. Hyperthermia is a method used to treat tumors by raising local or regional temperature through the use of controlled heat sources. The treatment might be applied in combination with other approaches in order to allow greater accumulation of drugs in the heated region and may increase efficacy and decrease side effects [6,7]. Therefore, the purpose of this review was to describe the most common thermosensitive nanocarriers used for tumor-specific drug release. In addition, we reviewed the most recent preclinical studies (2009–2019) involving thermosensitive systems associated with hyperthermia for the treatment of cancer.

## 2. Hyperthermia

Hyperthermia is a highly controlled method of heating of tumors, tissues, or systems to temperatures above the physiological temperature (37 °C) [8]. Heat induces physiological alterations in cells in a time-dependent and temperature-dependent manner [8]. Hyperthermia treatment in oncology was first described in 1898 by Frans Westermark, a gynecologist who obtained an excellent response in advanced cervical carcinomas by running hot water into an intracavitary spiral tube [9]. It has subsequently been shown that there is a tumor-selective effect of hyperthermia at temperatures between 40 °C and 43 °C [9]. There are three zones impacted by hyperthermia: Central, peripheral, and outer [10]. The central zone is the direct and immediate site of heat transfer and cells generally die of necrosis. As the heat disseminates to the peripheral and outer zones, the impacts are more indirect and associated with apoptotic pathways and influenced by altered microenvironment. Hyperthermia leads to membrane fluidity and dysfunction through alterations in transport proteins, ion channels, receptors, and lipids [10]. Within the cell, hyperthermia denatures proteins, altering their structure and function. This process can be reversible if proteins recover through refolding pathways. Hypoxia in the core of tumors has shown to be a clinical challenge due to the low pH levels and poor blood supply [11]. The tumor region is known to have acidic pH, altered vasculature, and poor lymphatic drainage. These features can be used in favor of cancer treatment through enhanced permeability and retention (EPR) effect [12]. In combination with warming, hypoxia conditions render tumors more sensitive to hyperthermia, especially in areas with low perfusion. Thus, hyperthermia may induce direct cytotoxicity, as well as lead to selective destruction of tumor cells in hypoxic and, consequently, acidic parts of solid tumors [6,10,13].

Ablation and mild hyperthermia are two standard methods to achieve hyperthermia clinically (Figure 1). Ablation refers to a short burst of high temperatures (> 50 °C, for 10 min), whereas mild hyperthermia is achieved by applying lower temperatures for long periods (39–42 °C, for approximately 60 min) [11,14]. The physiological response to these strategies is distinct. During ablation, hyperthermia may provoke denaturation and coagulation of cellular proteins, rapidly destroying the cells inside the target tissue [15]. Although thermal ablation can effectively destroy tumor tissue, a major limitation is the difficulty of heating large tumors, since the entire tumor cannot reach an adequate temperature for coagulation and necrosis [16].

In contrast, a mild heat treatment may induce numerous changes in cellular and molecular physiology, and it has not been associated with any toxicity. Several targets within the cell may be affected due to an increase in temperatures, including membranes, cytoskeleton, and synthesis of macromolecules [17,18]. Mild hyperthermia may also trigger changes in perfusion and oxygenation, along with inhibition of DNA repair mechanisms. Additionally, there is evidence of immune stimulation and the development of systemic immune responses [19]. Hyperthermia causes biochemical changes due to a thermal shock within the cell, including a reduction in cell division and an increase in sensitivity to ionizing radiation therapy. It can also improve blood flow to the heated area, doubling the perfusion in the tumors [11], which intensifies drug delivery and prevents cells from repairing the damage induced during the radiation session [15,17,20].

There have also been reports of increased blood flow in most human tumors under conditions of hyperthermia, even hours after treatment induction. This increase in blood perfusion occurs preferentially at the onset of tumor warming, through an improvement of the microcirculation [13,21]. This may play a pivotal role in altering hypoxic conditions which are known to contribute to radioresistance. Specifically, hypothermia treatment coupled with radiation leads to radiosensitization [19]. Thermal damage can alter the protein structure of non-histone nuclear proteins, which are especially sensitive to heat, causing their unfolding and aggregation due to the exposure of hydrophobic groups, and subsequent association with the nuclear matrix. Consequently, functions dependent on the nuclear matrix, such as transcription, replication, or DNA repair, are compromised [22].

Issels and co-workers proposed six important characteristics as “hyperthermia marks”: (1) block cell survival, (2) induce a cellular response to stress (3) modulate the immune response, (4) prevent DNA repair, (5) alter tumor microenvironment, and (6) radiation and chemotherapy sensitization [23]. Specifically, the use of extreme hyperthermia (≥ 50 °C) causes endothelial damage, diminishing blood flow, and enhances hypoxia and acidosis, while moderate hyperthermia (≤ 42 °C) improves the blood flow of the tumor and can therefore act as a radiosensitizer, enhancing oxygenation, and as a chemosensitizer in the tumor environment [8].

Leveraging hyperthermia treatment strategies combined with radiation and chemotherapy is an emerging therapeutic platform with promising traction [24,25]. Combinations of hyperthermia with conventional therapies have the significant advantage of yielding lower doses of chemotherapy and radiation, leading to more effective treatment with fewer side effects and reduced resistance of cancer cells [7,17]. Some studies have shown a synergistic effect when hyperthermia and radiation are coupled, which is called “thermal radiosensitization”. This effect induces an increase in cell death even at lower temperatures and seems more pronounced at the S-phase of cell division, whose cells are normally resistant to the isolated radiation [26]. Furthermore, other studies show that heat modifies the cytotoxicity of many chemotherapeutic agents, and this process is known as "thermal chemosensitization" [6,11,17,27,28]. It has been demonstrated that alkylating agents (cyclophosphamide and ifosfamide), platinum compounds (cisplatin), and nitrosoureas (carmustine (BCNU) and lomustine (CCNU)) have a more potent cytotoxic effect if the temperature is increased from 37 to 40 °C [29]. Doxorubicin (DOX) or bleomycin was shown to induce enhanced cell death at higher temperatures, 42.5 °C [11,29]. Table 1 shows some clinical trials using the combination of hyperthermia and chemotherapy or radiotherapy.

### Types of Hyperthermia Treatments

Achieving temperatures above the physiological temperature of 37.5 °C at a defined target is a great challenge and depends on the size of the tumor and depth of target tissues [36]. In general, elevated temperatures are achieved using external devices, which transfer the energy to the tissue, resulting in clinically effective thermal doses without provoking intolerable tissue temperatures [15]. There are two basic strategies to achieve hyperthermia; the first uses high energy waves from an external machine, and the second uses a thin needle or probe inserted directly into the target tissue guided by ultrasound, magnetic resonance imaging (MRI), or computed tomography (CT). According to the National Cancer Institute, hyperthermia can be applied locally, regionally, or systemically.

Local hyperthermia is appropriate for relatively small tumors (≤ 3 to 5–6 cm), located superficially or inside a body cavity, such as the rectum or esophagus [6]. Using a probe, heat is applied to a small area and there are a variety of approaches depending on the location of the tumor: External, endocavitary, and interstitial. External measures are used to treat more superficial tumors. External applicators are positioned around the region, and energy is specifically directed to the tumor [37]. This heating process can be performed by using applicators, which emit mainly microwaves or radio waves [36]. The endocavitary method can be used to treat tumors within or near body cavities. The probes are placed into the cavity and inserted into the tumor in order to provide energy, heating the area directly [37]. Interstitial techniques allow higher temperatures than external techniques. They are performed under anesthesia and can be used to treat deeper tumors, such as brain tumors. Imaging techniques can also be implemented to ensure that the probe is correctly positioned within the tumor. Various types of applicators are available, including microwave antennas, radiofrequency electrodes, ultrasonic transducers, and laser fibers [36].

In regional or isolated perfusion hyperthermia, some of the patient’s blood is removed, heated, and then pumped (perfused) back into the limb or organ, and anticancer drugs are commonly given during this treatment [6]. Thus, large areas of the body undergo heating, such as tissues, body cavities, organs, or limbs [37], in order to reach deeper tumors such as the pelvis or abdomen. Treatment monitoring can be performed by MRI by indicating temperature and perfusion [36]. However, regional hyperthermia is more complicated than local heating, due to variability in the physical and physiological properties of different tissues. Thus, further refinement, temperature measurement, and quality assurance are required. This method has been approved in phase III studies using slightly invasive thermometry catheters with no significant side effects [6].

Finally, whole body hyperthermia is commonly applied, isolated or in combination with chemotherapy, for the treatment of metastatic disease [19]. The principal behind whole body hyperthermia is to achieve “fever range” (107 °F or 41 °C) as an effective means to activate immune cells. It can be performed by several techniques that increase body temperature, including the use of thermal chambers (similar to large incubators), warm water immersion tanks, or warming blankets [37]. These procedures can only be executed with deep analgesia and sedation or general anesthesia. Currently, only radiation systems are in clinical use, with preheating times of 60–90 min (above 37.5 °C) [36,38].

## 3. Heating Modalities Used to Induce Hyperthermia

Different types of energy can be used to apply heat, including ultrasound, radiofrequency, and microwave [8,17]. Ultrasound involves the propagation of sound waves at a frequency of 2 to 20 MHz through the tissue that leads to their absorption and results in regional elevated temperatures [38,39]. The high-intensity focused ultrasound (HIFU) technique allows both ablation and moderate hyperthermia of the tumor. This technique concentrates ultrasound waves from outside the body into a target deep beneath the intact skin, without affecting nearby tissue [40]. The latter also allows the targeted delivery of drugs. Thermometry is the use of MRI to measure changes of temperature within a magnetic field. Mapping of temperature change in vivo, induced by HIFU, is possible employing MRI–HIFU, providing, therefore, accurate tissue heating over an extended time [16]. One advantage of this modality is that multiple exposures side-by-side allow more effective treatments to larger volume tumors. This methodology is clinically approved for the treatment of non-malignant tumors of the uterus and bone metastasis and is under current investigation for the treatment of prostate cancer and various other cancer indications [40].

In order to heat large tumors in depth, radiofrequency (RF) fields in the range of 10 to 120 MHz are generally used with long wavelengths, compared to body dimensions, and thus depositing energy over a considerable region [38]. They consist of pairs of dipole antennas around the patient. Each pair of antennas can be controlled in phase, amplitude, frequency, and electric field to focus the heat on the tumor area with an accuracy of a few centimeters. This method generates high frequency alternating currents that cause rapid oscillations to nearby cells to yield frictional heat. Treatment monitoring can be achieved with MRI, which may characterize temperature change as well as perfusion [40]. In the types of equipment used in the RF technique, the radiofrequency is emitted by a generator and passes through the tumor tissue from the tip of a needle-like probe (active electrode) that is inserted into the tumor. The radiofrequency current transmitted by the active electrode traverses the tumor toward a dispersive electrode, which is usually firmly attached behind the patient’s right shoulder or thigh. Active and dispersive electrodes have different dimensions, which results in a difference in current density between them. This situation contributes so that the energy generated at the tip of the active electrode causes agitation of the ions present in the tumor tissue. The tissue ions are agitated as they attempt to follow the changes in direction of alternating electric current. Finally, this agitation is transformed into heat that causes the destruction of tumor tissue by induction of necrosis [41]. Several clinical trials support the viability and efficacy of this method in combination with standard chemotherapy or radiotherapy for the treatment of solid tumors [26,31]. These studies provide preliminary evidence that hyperthermia, in addition to standard radiotherapy, may be especially useful in locally advanced cervical tumors. However, studies of larger numbers of patients are needed.

Another non-invasive technique and effective strategy for heating tumors utilizes microwave energy, which predominantly use antennas that run at 434, 915, and 2450 MHz [38]. A microwave system includes the antenna and a non-contact temperature sensor that sweeps a predetermined path over the surface of the tissue to be treated. Since the body tissues contain a high-water content, when they are irradiated with microwave energy, the temperature rises in the tissue due to the transfer of microwave energy into heat. This method generates an oscillating electromagnetic field that uses ions and dipoles to align and rotate, causing friction that heats the tissue within an effective range of 3 cm. It has been widely used in patients suffering from prostate or breast cancer [38,42]. In conclusion, the results validate the capability of the proposed technique in focusing power at the exact location and volume of the tumor, but there is still room for further technological improvements.

Laser-induced hyperthermia is a promising method for the treatment of superficial malignant tumors [43]. During this procedure, light energy is absorbed into the tissue, starting from the surface of the skin. Consequently, the maximum thermal energy produced is set in the skin and in the layers immediately below its surface (dermis and subcutaneous layers). The epidermis contains melanin, which is characterized by high light absorption, allowing thermal laser [43,44]. In conclusion, the correct choice of the parameters used for treatment and planning should be studied. As a result, this technique is recommended for (and limited to) soft thermal treatment of superficial tumors.

The use of magnetic fluid hyperthermia destroys cancer cells by heating tissue impregnated with ferrofluid and a magnetic field, causing minimal damage to the surrounding healthy tissue [45]. While the potency of the magnetic field should be sufficient to induce hyperthermia, it is also limited by the human capacity to stand safely. The ferrofluid material used for hyperthermia should be non-toxic and also be able to provide adequate heating [45]. Magnetic nanoparticles (MNPs) have been extensively used as agents of magnetic fluid hyperthermia. Optimizing MNP-based nanoagents for early diagnosis and efficient therapeutics associated with hyperthermia is required. With a multidisciplinary approach leveraging chemistry, physics, biology, and pharmaceutical sciences, it can postulated that MNP agents will achieve the high sensitivity and efficacy for clinical diagnostics and therapeutics in the future [46].

The development of superparamagnetic iron oxide nanoparticles (SPIONs) for diagnosis and/or bimodal cancer therapy, using magnetic hyperthermia and radionuclides, has grown considerably over recent years [47,48,49,50]. Mokhodoeva and collaborators described the incorporation of 223Ra (radium-223), the first clinically approved alpha-emitter, into SPIONs. The authors highlighted that the application of these SPIONs for theranostic purposes would benefit effective targeting and dose-delivery to target regions [48]. Another study published in the same year reported the incorporation of three radionuclides, technetium-99 (99mTc), yttrium-90 (90Y), and lutetium-177 (177Lu), into SPIONs. It was postulated that the nanoparticles with high energy beta emitters 90Y and 177Lu can be a promising system for magnetic hyperthermia and radionuclide therapy. On the other hand, SPIONs with 99mTc could be used for diagnostic imaging purposes [49]. Studies by Pospisilova and collaborators reported the incorporation of radionuclide 59Fe (iron-59) into SPIONs [50]. Separate studies showed the incorporation of 59Fe and 111In (índium-111) into SPIONs [47]. This study was the first to describe the in vivo integrity of radiolabeled SPIONs intended for use with nuclear medical imaging technologies. It is well known that the application of external magnetic fields has the potential to influence SPION’s physiological biodistribution and concentrate them to a specific body region [51]. In this context, a recent study demonstrated that the micro positron emission tomography (PET) and the micro CT were able to provide a multimodal three-dimensional (3D) data set, about biodistribution of SPIONs radiolabeled with 18F-2-fluoro-2-deoxyglucose (18FDG), following intravenous administration in a small animal model, with and without the application of a permanent magnet onto the skin [52].

## 4. Thermosensitive Systems for Cancer Treatment

Drug targeting systems have been developed specifically for the desired target, which may increase the effectiveness of therapeutic agents and delay the development of resistance. Given that the bioavailability of drugs at the cancer cells is very important, a nanocarrier system incorporated with temperature would be useful to overcome some of the systemic and intracellular delivery barriers. Therefore, the advancement in material science has led to the design of a variety of materials, which are used for the development of thermo-responsive nanocarrier systems [53]. Nanoparticle-based hyperthermia strategies have been considered promising nanocarriers for antitumoral drug delivery systems and their development has increased considerably in the last few decades [3,54].

These thermosensitive nanocarriers stand out for their stability at physiological temperature; however, they quickly release the drug into pre-selected sites in response to changes in the local temperature [4,55]. The use of these systems has assumed an important role as an alternative to improve the biodistribution profile of drugs, since they can increase the intratumoral drug accumulation and reduce systemic toxicity [56]. The literature is vast in advances in synthesis, characterization, and application of different thermosensitive nanoparticles. Although they can be composed of different kinds of materials, including biocompatible polymers, lipids, and self-assembling amphiphilic micelles, most of them have shown a polymeric composition [53,57]. In this review, we describe the most common nanocarriers used for a thermo-responsive antitumor drug release. In addition, we review the most recent preclinical studies (2009–2019) involving thermosensitive systems associated with hyperthermia for the treatment of cancer.

### 4.1. Polymeric Nanocarriers

The thermo-responsive propriety of polymers is based on their response to thermal stimuli due to chain modification [55]. In thermo-responsive nanosystems, a temperature-sensitive polymer is used as a fabrication material, which displays a critical solution temperature at which the polymer system undergoes a phase change within a temperature, thus allowing the delivery system to release the cargo upon the changes in temperatures [55]. The temperature is dependent on the interactions between polymer–polymer or polymer–solvent. At a critical solution temperature, the polymeric solution undergoes separation into two phases [58,59]. Many challenges are involved in the development of polymers with transition temperatures within the required temperature range of 38–40 °C, since the phase transition temperature is strongly influenced by polymer concentration, pH, ionic strength, as well as the presence of specific molecules or ions in solution [60,61]. These thermosensitive polymers are characterized by a reversible phase transition. At low temperatures, they are water-soluble, while above a critical transition temperature, they change their conformation and the hydrophilic/hydrophobic balance, which provokes phase separation and presence of aggregates [62,63]. Temperature-responsive amphiphilic polymers often have thermosensitive hydrophilic segments and a suitable hydrophobic segment in their structure. An example of this type of polymer is the family of poly-polymers (N-substituted acrylamide), as the poly(N-isopoprylacrylamide) (pNIPAAm) [63,64]. Hruby and collaborators described systems based on pNIPAAms with isotopically labeled end groups (L-tyrosinamide or diethyltriaminepentaacetic acid) designed for local radiotherapy [65]. The binding capacity for radionuclides and chemical stability was demonstrated using iodine-125 (125I) and 90Y labeled polymers [65]. The same group performed the radiolabeling of a thermo-responsive system with copper-64 (64Cu) [66]. The system is based on the copolymers of N-isopropylmethacrylamide with three different monomers containing hydrophobic n-alkyls groups of different sizes bound to the methacrylamide unit by a hydrolytically labile hydrazone bond [66]. Another study from the same group described a new thermo-responsive polymeric system, based on pNIPAAms, for local chemoradiotherapy using DOX and the radionuclide 125I [67].

Thermo-responsive polymers used in the biomedical field can get in contact with ionizing radiation when used as carriers of radiopharmaceuticals or during radiation sterilization procedures. In this context, a recent review described the effect of ionizing radiation on the physicochemical and phase separation properties of some thermo-responsive polymers [68]. In this study, the poly(2-isopropyl-2-oxazoline-co-2-n-butyl-2-oxazoline) (POX) was presented as the most suitable polymer for the development of delivery systems that can be exposed to radiation. On the other hand, the polymer poly(N-vinylcaprolactam) (PVCL) was considered the least suitable for this purpose. PNIPAAms and poly[N-(2,2-difluoroethyl)acrylamide (DFP)] are considered suitable only for low radiation exposures [68].

It has been also reported that the copolymerization causes alteration of the phase transition temperature [61]. Thus, an increase in hydrophobic monomers or an increase in molecular weight may result in a decrease of the critical solution temperature [69]. On the other hand, the incorporation of hydrophilic monomers forms hydrogen bonds with thermosensitive monomers, increasing the critical solution temperature point [69,70].

Another class of thermosensitive polymers is Pluronic. They are triblock copolymers of polypropylene oxide (PPO) middle blocks flanked by polyethylene glycol (PEG) blocks (PEG-b-PPO-b-PEG) [71]. Pluronic F-127 or poloxamer 407 (Pluronic^®^, St. Louis, MO, USA) is a commercially available polyoxyethylene–propylene copolymer, which contains around 70% ethylene oxide, responsible for its hydrophilicity [72]. It has been widely used due to its thermosensitive character, low toxicity, and high solubilizing capacity of poorly water-soluble drugs, such as the various chemotherapeutic drugs, and newly developed anticancer compounds have high lipophilicity [71,72].

Among the wide variety of thermosensitive nanosystems, micelles, polymer nanoparticles, and polymersomes are the most frequently studied.

#### 4.1.1. Polymer Micelles

Thermo-responsive polymeric micelles are formulated through a self-assembly process using amphiphilic block copolymers that spontaneously assemble into a core–shell structure in an aqueous environment above the critical micelle concentration (CMC) [55]. A low CMC is required to guarantee the stability of micelles after injection, since the bloodstream dilution might result in the disruption of self-assembled particles [55]. The hydrophobic core of micelles allows carrying cargo of a chemotherapeutic agent, normally a water-insoluble drug, while the temperature-responsive hydrophilic shell polymer favors the delivering of the drug at a specific site [73].

Many thermosensitive polymer-based micelles have been described in the literature, among which pNIPAAm, Pluronics, and poly(hydroxypropyl methacrylamide-lactate) (p(HPMAm-Lacn)) are the most frequently studied, and some drug-loaded formulations based on thermosensitive polymers have reached clinical trials [74]. They are water-soluble polymers below these phase transition temperatures and water-insoluble polymers above this temperature. Although PNIPAM-based systems were the focus of various studies as drug nanocarriers, many did not present satisfactory release results, because pNIPAAm is not suitable for in vivo applications as its phase transition temperature is below body temperature [73]. Polymers synthesized using oligoethyleneglycol (OEG) monomers are attractive alternatives to PNIPAAm, with several advantages; the polymers can be synthesized by controlled radical polymerization, they have reversible phase transitions, and they form effective shields against protein adsorption [75]. However, it remains a challenge due to the molecular weight and the slow degradation profile of the polymers exhibiting the phase transition in the desired range. One strategy to overcome these limitations would be the development of thermosensitive nanostructures formed by phospholipids [75].

The preclinical study by Liu et al. is an example of a novel docetaxel-loaded micelle based on the biodegradable thermosensitive copolymer poly(N-isopropylacrylamide-co-acrylamide)-b-poly(DL-lactide) that demonstrated that hyperthermia increased the antitumor efficacy of these thermosensitive micelles [76]. This formulation was extended to evaluate the efficacy in BGC (human gastric carcinoma xenograft models). These studies showed that at the same dose level of docetaxel/paclitaxel, hyperthermia greatly enhanced the antitumor effect through growth inhibition of more than 80%. The present results suggest that poly(IPAAm-co-AAm)-b-PDLLA micelles could be a clinically useful chemotherapeutic formulation and merit further research to evaluate the feasibility of clinical application [77].

In another preclinical study, Chen et al. investigated thermosensitive folate micelles composed of P(FAA-NIPA-co-AAm-co-ODA) and P(FPA-NIPA-co-AAm-co-ODA) containing paclitaxel associated with hyperthermia in the treatment of lung cancer [75]. The encapsulation in P-NIPAm micelles and local heating drastically increased the accumulation of paclitaxel at tumor sites, local drug concentration was greatly enhanced, and thus the treatment effect improved, demonstrating that these micelles are a promising candidate for high treatment efficacy in tumor therapy [75]. Table 2 summarizes the most recent preclinical studies using hyperthermia and thermosensitive micelles.

#### 4.1.2. Core–Shell Nanoparticles

Nanoparticles with inorganic cores and thermosensitive polymer shells are an interesting class of composite materials. They combine the properties of both the core and the shell [58]. Thermo-responsive core–shell nanoparticles are also studied for drug delivery applications [58]. In general, the thermo-responsive molecule is located on the surface and the core can be constituted by either a hard metallic (gold, magnetic) or a soft (dendrimers, chitosan, silica, nanogels) nanoparticle [2,75]. The rationale is to combine polymers and superparamagnetic nanoparticles to trigger drug release. Their incorporation into biopolymer coatings enables the preparation of magnetic field-responsive, biocompatible nanoparticles that are well dispersed in aqueous media [78]. Therefore, thermo-responsive nanomaterials of greater interest are those that respond to other stimuli besides temperature. With this, additional stimuli-responsive characteristics (magnetic field, ultrasound, light, and heat) can be added to the thermal response [53]. In order for the nanomaterials to exhibit such a response, either iron oxide or gold can be incorporated in the form of nanoparticles to the polymer matrix [79]. Iron oxide-based magnetic nanoparticles are extensively investigated in nanomedicine for their biocompatibility, their contrast agent properties, and their ability to generate heat when submitted to an alternating magnetic field [80,81]. Wang et al. designed a multiple magnetic hyperthermia-mediated release system for combination therapy using an injectable, biodegradable, and thermosensitive polymeric hydrogel [82]. Wang et al. noted that the therapeutic effect of the combination therapy in vivo significantly reduced the tumors and was dependent on the number of cycles of hyperthermia [82].

Li et al. synthesized magnetothermal responsive nanocarriers of DOX with targeting molecules, which allows the antitumor drug to locate tumor sites efficiently [83]. Magnetic nanoparticles composed of manganese–zinc (Mn–Zn) could target hepatoma cells actively and improved the drug concentration in the tumor sites. The shell of the nanoparticles was composed of an amphiphilic and thermosensitive copolymer to which DOX was associated. Therefore, an external alternating magnetic field elevated the temperature of the thermomagnetic particles, resulting in structural changes in the thermosensitive copolymer, thereby releasing DOX [83]. In another study, Shen et al. synthesized a thermo-responsive system, which encapsulated magnetic iron oxide nanoparticles and 5-fluorouracil using PNIPAM polymer. The results showed great potential in drug delivery and cancer therapy [84]. Other applications of hyperthermia-associated magnetic nanoparticles are described in Section 5 of this article.

Hydrogels have a cross-linked three-dimensional structure and are composed of hydrophilic polymers. The presence of hydrophilic groups such as amino (–NH2), carboxylic (–COOH), and sulfate (–SO3H) groups provides the ability to mimic natural living tissues [85]. Moderate temperature increases alter the interaction between the hydrophilic and hydrophobic segments in the polymer, inducing a change in network solubility, causing the sol–gel phase transition. Thus, the polymer is soluble below a certain critical temperature and insoluble when it is above this temperature [86]. The most common release mechanism of thermosensitive hydrogels occurs through passive diffusion, and different biotherapeutic molecules which are entrapped in the gel matrix can diffuse freely depending on the size of the mesh of this matrix. The hydrogels most studied are also based on pNiPAAm [87]. Copolymerization of pNiPAAm with more hydrophilic monomers increases polymer hydrophilicity, and stronger polymer–water interactions allow changes in the critical system temperature value. [88,89]. This copolymerization can be performed with polysaccharides, such as cellulose or chitosan, which are biodegradable and non-toxic compared to other natural polymers [90]. Turabbe and collaborators designed a thermosensitive hydrogel from pluronic F127 (PF127) and N,N,N-trimethyl chitosan in order to deliver DOX for glioblastoma multiforme [91]. This hydrogel showed a sustained release of tumor-suppressed DOX, suggesting it as a potential candidate for an anticancer chemotherapeutic delivery system for brain tumors [91]. In the same year, Pesoa and co-workers prepared paclitaxel-loaded poly(lactide-co-glycolide) microparticles contained in a chitosan-sensitive gelling solution. This system allowed a more sustained and controlled administration of paclitaxel, leading to a long-term effect at the site of action. The formulation showed a tumor volume inhibition of 63.0% compared to the control group [92].

### 4.2. Liposomes

Liposomes are spherical vesicles composed of a lipid bilayer, usually consisting of phospholipids, which involve an aqueous core [2,93]. These systems allow encapsulating the hydrophilic or lipophilic drugs into the aqueous core or the lipid membrane, respectively [93]. The first paper on the concept of using temperature-sensitive liposomes to achieve enhanced delivery of anticancer agents was published in 1978. After decades of research, several types of thermosensitive liposomes were constructed, including the commercially available liposome known as ThermoDox^®^ [29]. Thermosensitive liposomes are designed to provide the drug release at mild temperatures due to the phase transition temperature (Tc) of the lipid composition [94]. Tc is defined as the temperature required to induce a change in the lipid physical state from the ordered gel phase, where the hydrocarbon chains are fully extended and closely packed, to the disordered liquid crystalline phase, where the hydrocarbon chains are randomly oriented and fluid. This parameter consists of altering of the hydrocarbon chain from a gel phase to a liquid crystal phase (Figure 2), which results in the disorganization of the bilayer and consequent drug release. In the gel phase, the chains are ordered allowing the formation of nanostructures. In the liquid crystal phase, the mobility of the molecules in the bilayer increases gradually and they become more disordered, and cannot maintain the liposomal structure. Thus, it is important to choose a phospholipid mixture with a suitable phase transition temperature within a hyperthermia range recommended for clinical application [24,95].

Traditional thermosensitive liposomes are based on dipalmitoyl phosphocholine (DPPC), a phospholipid with 16-carbon saturated fatty acid chains and a phase transition temperature of about 41 °C [96]. Supplementation of DPPC with other lipids, predominantly distearoyl phosphocholine (DSPC) and hydrogenated soy phosphocholine (HSPC), has an effect of improving the rate of drugs released [97]. Although the DPPC supplementation with other lipids can increase the permeability of the membrane, the inclusion of lipids with carbon chains longer than DPPC may have undesirable effects on the phase transition behavior [29,96,97]. In 1999, Anyarambhatla and Needham launched the idea of incorporating lysolipids into PEGylated DPPC membranes of traditional thermosensitive liposomes in order to reduce the phase transition temperature and promote rapid drug release [29]. The incorporation of the lysolipid monopalmitoyl phosphocholine (MPPC) reduced the phase transition of traditional thermosensitive formulations from 43 to 39–40 °C and provided a rapid release upon heating. This formulation, composed of DPPC, 1-stearoyl-2-hydroxy-sn-glycero-3-phosphatidylcholine (MSPC), and DSPE–PEG (in the molar ratio of 86:10:4) and containing encapsulated DOX, was marketed under the name ThermoDox^®^. This thermosensitive liposome exhibits a phase transition temperature of 41.5 °C and is the most advanced and effective temperature-activated nanocarrier available [29,94,96].

Thermosensitive liposomes composed of lysophospholipids constitute an important class of these nanosystems responsive to temperature, since the incorporation of a small quantity of lipids results in changes in the curvature of the phospholipids and leads to reduction in the membrane’s ability to act as a barrier [98]. However, further destabilization of the bilayer may cause premature drug leakage at physiological temperature, invalidating the clinical success of thermosensitive vesicles containing lysophospholipids [99].

Some studies have focused on the design of the generation of thermosensitive liposome that combines targeting and triggered drug release [94]. Various ligands such as folate, antibodies, receptors, and peptides have been conjugated to thermosensitive liposome [94]. Another approach is to sensitize non-thermosensitive liposomes by functionalizing them with temperature-responsive polymers that disrupt the membrane in response to heating [97].

There have recently been many preclinical studies involving this type of formulation. A DPPG2 liposome developed by Limeer et al. showed a higher release of gemcitabine above 40 °C. The plasma half-life of the drug was greatly increased from 0.07 to 2.59 h by using the thermosensitive liposome. Treatment of tumor BN175 with encapsulated gemcitabine associated with hyperthermia showed significant improvement in tumor growth inhibition compared to free gemcitabine [100]. Peller et al. intravenously administered doxorubicin thermosensitive liposome (DOX–TSL) associated with tumor heating above 40 °C for 1 h using laser light [101]. The result was a highly selective uptake of DOX, and the concentration of DOX in the heated tumor tissue compared to the unheated tumor showed an almost 10-fold increase [101]. Willerding et al. performed another study involving the encapsulation of DOX [102]. This group used different methods of hyperthermia and showed through preclinical studies that, regardless of the hyperthermia methods used, higher levels of DOX were found in tumors. This shows the increase in the concentration of drugs in the tumor after application of hyperthermia [102].

In vivo studies by Wang and co-workers, after three injections of a thermosensitive liposome containing paclitaxel in the lung tumor model, showed suppression of tumor growth compared to non-temperature-sensitive liposome and free drug [82]. Lokerse et al., in 2017, examined the potency of the combined therapy of a thermosensitive liposome with hyperthermia in two experimental models of human breast cancer-bearing mice [103]. Both cell lines showed improved in vitro chemosensitivity and increased uptake of DOX in the presence of hyperthermia [103]. Table 3 summarizes other examples of the most recent preclinical studies using hyperthermia and liposomes.

## 5. MRI-Guided Thermometry: A Strategy for Real-Time Monitoring and Control of Drug Release

Recently, there has been an increase in interest in exploring the synergy between imaging and drug delivery. This interest increased in vivo studies reporting the release of a drug from nanocarrier [104,105]. Among the available imaging modalities, MRI is an excellent candidate because of the excellent space-time resolution, the possibility of obtaining deep tissue/organ images, and the vast portfolio of available probes and contrast generation modalities [104]. A practical approach to visualize the release of the drug from liposomes is to encapsulate a hydrophilic paramagnetic agent (based on Gd^3+^ or Mn^2+^ ions) in the aqueous nanovesicle interior. After entrapment, the MR contrast is "silenced" and its activity is recovered when the agent is released [105].

A thermosensitive liposome formulation co-encapsulating DOX and a contrast agent was developed by Negussie et al. [106]. The stability of the technique, visualization ability, and content monitoring by magnetic resonance imaging associated with high-intensity focused ultrasound (MRI–HIFU) suggested that this technique combined with thermosensitive liposome could enable real-time monitoring and spatial control of drug release [106]. Temperature-sensitive liposomes (TSLs) co-encapsulating DOX and 250 mM Gadolinium [Gd (HPDO3A) (H2O)], a clinically approved T_1_-weighted MRI contrast agent, were also evaluated for delivering HIFU-mediated drugs under MRI [107]. A good correlation between the uptake of DOX and the concentration of gadolinium in the tumor was found, implying that the in vivo release of DOX from TSLs can be probed in situ with the time of longitudinal relaxation of the co-encapsulated MRI contrast agent. Anatomical MRI images of the tumor and surrounding mouse muscle were acquired before the administration of TSLs and directly after the first and second local hyperthermia treatment (Figure 3). It was possible to observe the difference of the tumors with MRI, before the administration of the TSLs and after the administration of TSLs in combination with mild hyperthermia, which induced significant changes in the tumor [107].

Staruch in 2015 evaluated the efficacy of DOX encapsulated in a thermosensitive liposome in combination with mild hyperthermia produced by high-intensity focused magnetic resonance-guided ultrasound (Figure 4) [108]. The authors concluded that tumors treated with a single thermosensitive liposome infusion during MRI–HIFU light hyperthermia reduced tumor growth compared to tumors treated with liposome without hyperthermia [108].

Hijnen et al., also investigated the use of MRI–HIFU with the intravascular release of DOX encapsulated in TSLs in a preclinical setting using a rhabdomyosarcoma mouse tumor model [109]. All HIFU heating strategies combined with TSL resulted in increased DOX cell uptake in the interstitial space and a significant increase in drug concentrations in the tumor compared to a free DOX treatment [109].

## 6. Clinical Trials

The first thermosensitive formulation of liposomes in the clinical trial phase is ThermoDox^®^ (Lawrenceville, GA, USA) [24]. This formulation is composed of DPPC/MSPC/DSPE-PEG_2000_ (86.5/9.7/3.8) with a concentration of 2 mg/mL Doxorubicin. DPPC presents a phase transition temperature of DPPC near 42 °C and the presence of MSPC provokes a reduction in this temperature. Thus, phase changes are easily reached by the use of mild hyperthermia [75]. ThermoDox^®^ is administered by the intravenous route in combination with radiofrequency ablation (RFA) for treatment of primary liver cancer, as well as recurring chest wall breast cancer. As the Phase I results in patients with hepatocellular carcinoma were promising, this combination of treatment was directly evaluated in Phase III. The initial Phase III results did not show any progression of survival with the combination (TermoDox^®^ plus RFA, Lawrenceville, GA, USA) compared to RFA alone. The reasons for the failure of the heating test were related to problems with the clinical trial design, few preclinical supportive data, and required improvements in heating control. However, an analysis of the patients’ subgroup who received RFA for at least 45 min showed an improvement in overall survival [24].

Another thermosensitive nanosystem under clinical investigation is a core–shell, a gold nanoparticle coated by polyethylene glycol (PEG), AuroLase™ (Houston, TX, USA) [110]. The multicenter, single-dose pilot study of AuroLase^TM^ therapy was applied in the treatment of patients with refractory and/or recurrent tumors of the head and neck. Three treatment groups of five patients each were enrolled and observed for six months following treatment. Each group received a single dose of this nanoparticle, followed by one or more interstitial illuminations with an 808 nm laser. This gold nanoparticle consists of a non-conducting silica core and acts as the exogenous absorber of the near-infrared laser energy delivered by the probe. A currently active trial uses AuroLase^TM^ as an imaging technology during focal ablation of prostate tissue using nanoparticle-directed laser irradiation. Since no active drug is used in this approach, AuroLase^TM^ is activated externally at the target site, thus avoiding any toxicity towards the normal cells [110,111].

## 7. Conclusions

In conclusion, mild hyperthermia is not associated with toxicity in contrast to radiotherapy and chemotherapy. However, hyperthermia has not been widely used in the clinic. The major hurdles in the clinical translation and widespread use of hyperthermia are largely centered on technical challenges and infrastructure. Contrary to most other nanoparticle approaches, thermosensitive liposomes can be easily employed for image-guided drug delivery, where the goal is to deliver the drug to a region identified by medical imaging. Based on the studies carried out in recent years, therapy with thermosensitive nanosystems associated with hyperthermia has been shown to overcome challenges faced by therapies based on free drugs or non-thermosensitive nanosystems. A large number of available approaches to generate these smart nanocarriers illustrates the versatility of the system and the great potential to be explored for future applications.

The key to future research and development of nanosystems associated with hyperthermia relies on the development of valid and non-invasive ways to promote and measure temperature, as well as to understand the variability in the individual physiological responses and extrapolating laboratory studies to field settings. Thus, more advanced preclinical studies are required, in addition to clinical studies, to prove the efficacy of these systems.

## Figures and Tables

**Figure 1 pharmaceuticals-12-00171-f001:**
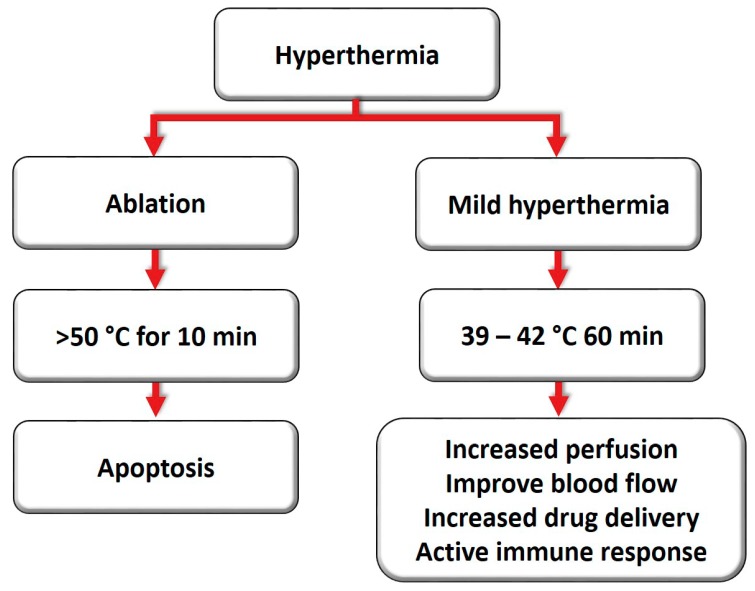
Ablation and mild hyperthermia induce distinct cell injury based on the intensity and duration.

**Figure 2 pharmaceuticals-12-00171-f002:**
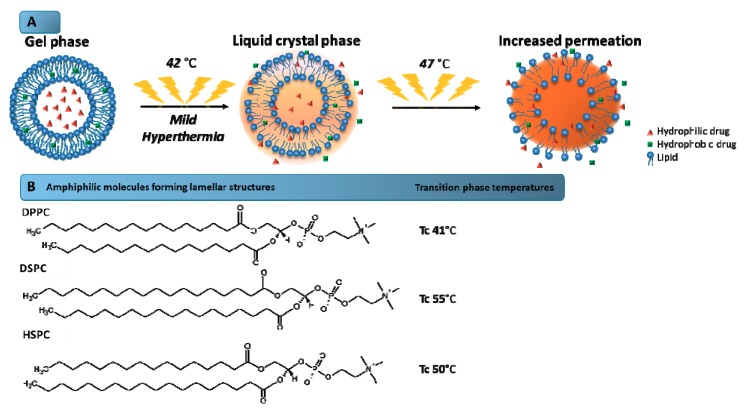
Mechanism of drug release from thermosensitive liposomes. (**A**) Schematic illustration of the mechanism of phase transition of the lipids that form the liposome bilayer. The increase of the temperature above the transition phase temperature (47 °C) leads to higher bilayer permeability, and consequently, the drug release is favored. (**B**) Amphiphilic molecules forming lamellar structures and their transition phase temperatures. Tc: Phase transition temperature.

**Figure 3 pharmaceuticals-12-00171-f003:**
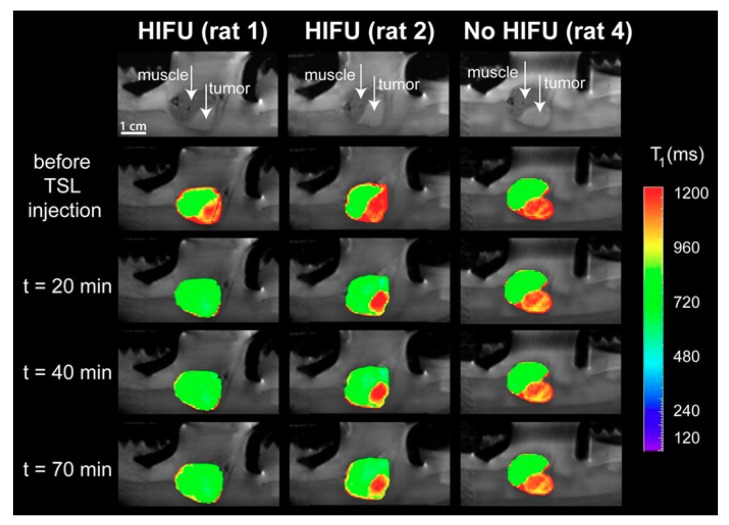
Anatomical MRI of tumor-bearing rats in the small animal HIFU setup (upper row) and T1 maps of the tumor and leg overlaid on the anatomical images at different time points: Before the TSL injection, after the first hyperthermia period (t = 20 min), after the second hyperthermia period (t = 40 min), and 70 min after TSL injection. Left: HIFU-treated tumor showing a large T1 response (rat 1); middle: HIFU-treated tumor showing a less sensitive response (rat 2); right: Untreated tumor (no HIFU, rat 4). Reproduced with permission from [107].

**Figure 4 pharmaceuticals-12-00171-f004:**
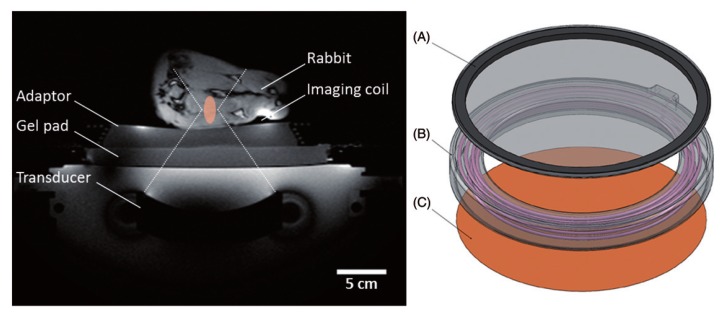
Experimental set-up for mild hyperthermia in rabbit V × 2 tumors using a clinical MRI–HIFU system. Axial survey image of a rabbit on top of a water-filled animal adaptor. A waterproofed receive-only imaging coil is fitted around the lower leg. The bottom film of the animal adaptor is coupled to the window of the clinical HIFU system by a gel pad; the HIFU transducer is in the oil bath below. Overlays indicate the relative size of the ultrasound beam path (dashed) and treatment cell (shaded). Right: Rendering of the animal adaptor designed for the clinical HIFU system. The detachable lid (**A**) is a polyimide film glued to an acrylic ring. The cylindrical water bath (**B**) is a 3D-printed shell that holds a volume of degassed water, which is heated by water pumped through a coiled channel printed into the walls of the cylinder. Polyimide film (**C**) forms the base. Reproduced with permission from [108].

**Table 1 pharmaceuticals-12-00171-t001:** Clinical studies using hyperthermia associated with radiotherapy or chemotherapy.

**Hyperthermia and Chemotherapy**				
**Drug**	**Target**	**Clinical Trial**	**Response**	**Reference**
Etoposide, ifosfamide, and doxorubicin	High-risk soft tissue sarcoma	Randomized phase III multicenter study	Local control, overall survival and response rate were better with HT+CT	[30]
Cisplatin and gemcitabine	Pancreatic carcinoma	Retrospective clinical study	HT+CT was well tolerated, and had an acceptable survival profile	[31]
Cisplatin and irinotecan	Malignant mesothelioma of the pleura	Case report	Increase in survival without any disease for more than 7 years	[32]
Nimustine	High-grade glioma	Phase I clinical study	HT+CT was tolerable in patients with relapse of high-grade gliomas	[33]
Trabectedin	Soft tissue sarcoma	Randomized clinical trial	High feasibility, no uncommon side effects, did not increase toxicity, and progression-free survival	[23]
**Hyperthermia and Radiotherapy**				
**Drug**	**Target**	**Clinical Trial**	**Response**	**Reference**
-	Cervical carcinoma	Long-term result after the 12-year segment	Local control and survival were better with HT+RT	[34]
-	Bladder, cervix, and rectum	Prospective, randomized, multicenter study	Complete response rates, local control, and survival were better with HT+RT	[26]
-	Breast cancer	Toxicity study	HT+RT was more effective for locally advanced or recurrent breast cancer than RT alone	[35]

HT: Hyperthermia; RT: Radiotherapy; CT: Chemotherapy.

**Table 2 pharmaceuticals-12-00171-t002:** Recent preclinical studies using hyperthermia and thermosensitive micelles.

Thermosensitive Micelles					
Composition	HT	Drug	Target	Response	Ref.
Poly(N-isopropylacrylamide-co-acrylamide)-b-poly(DL-lactide)	Water bath	Docetaxel	Lung cancer	Higher antitumor efficacy in mice treated with docetaxel-loaded micelles accompanied by hyperthermia	[76]
Poly(N-isopropylacrylamide-co-acrylamide)-b-poly(DL-lactide)	Water bath	Docetaxel and Paclitaxel	Gastric Cancer	Weight growth percentage inhibition of more than 80%	[77]
P(FAA-NIPA-co-AAm-co-ODA) and P(FPA-NIPA-co-AAm-co-ODA)	Water bath	Paclitaxel	Lung cancer	Increased accumulation of paclitaxel at tumor sites, local drug concentration was greatly enhanced	[75]

HT: Hyperthermia; P(FAA-NIPA-co-AAm-co-ODA): Poly(folate acrylic acid-N-Isopropylacrylamide-co-octadecyl acrylate); P(FPA-NIPA-co-AAm-co-ODA): Poly(folate-PEG acrylic acid-N-Isopropylacrylamide-co-octadecyl acrylate).

**Table 3 pharmaceuticals-12-00171-t003:** Recent preclinical studies using hyperthermia and thermosensitive liposomes.

Thermosensitive Liposomes					
Composition	HT	Drug	Target	Response	Ref.
DPPC/DSPC/DSPE–PEG_2000_ 70/25/5	Water bath	DOX	Breast cancer	Significant increase in tumor response to liposome and HT treatment	[103]
DPPC/DSPC/DPPG_2_ 50/20/30	Laser light	DOX	Soft tissue sarcoma	High selective DOX uptake and increase of DOX concentration in the heated tumor tissue	[101]
DPPC/DSPC/DPPG_2_ 50/20/30	Laser light	DOX	Soft tissue sarcoma	Effective DOX delivery by liposome found in the heated tumors in comparison with the non-heated tumors	[102]
DPPC/DSPC/DPPG_2_ 50/20/30	HIFU	Gemcitabine	Soft tissue sarcoma	Significant improvement in tumor growth delay	[100]
DPPC/MSPC/DSPE–PEG_2000_/DSPG 83/3/10/4	Water bath	Paclitaxel	Lung cancer	Tumor growth suppression, compared with non-temperature-sensitive liposome and free drug	[82]

HT: Hyperthermia; DPPC: 1,2-Dipalmitoyl-sn-glycero-3-phosphocholine; DSPC: 1,2-Distearoyl-sn-glycero-3-phosphocholine; DSPE: 1,2-Distearoyl-sn-glycero-3-phosphoethanolamine; DPPG2: 1,2-Dipalmitoyl-sn-glycero-3-phospho-rac-glycerol; MSPC: 1-Myristoyl-2-stearoyl-sn-glycero-3–phosphocholine; PEG: Polyethylene glycol.

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
