# Peer review of "Thermosensitive Nanosystems Associated with Hyperthermia for Cancer Treatment"

_pharmaceuticals, 2019, doi:10.3390/ph12040171_

Round 1
Reviewer 1 Report
The manuscript entitled "Thermosensitive nanosystems associated with hyperthermia for cancer treatment: a systematic review" by Gomes et al. reviews on hyperthermia and associated thermosensitive nanosystems for the treatment of cancer. Following are the major concerns regarding the manuscript:
The title includes "Systematic review" but the manuscript doesn't seem to be of that kind. No reference numbers are visible within the manuscript. English needs to be improved throughout the manuscript. There is no good story flow for the manuscript. There are a lot of repetitive information within the same sections of the manuscript. Extensive evaluations of the published thermosensitive systems for cancer treatment is missing.
Reviewer 2 Report
The paper “Thermosensitive nanosystems associated with hyperthermia for cancer treatment: a systematic review” is a review on thermosensitive system associated with moderate hyperthermia. It is an interesting subject, however there are many flaws in the manuscript that need to be changed for publication in Pharmaceuticals.
In the introduction (paragraph 1), the objective of the review is the description of the thermosensitive nanosystems used in preclinical and clinical studies associated with moderate hyperthermia. In the final part of paragraph 4, the objective of the revision becomes the description of the most common nanocarriers used for a thermo-responsive antitumor drug release. Perhaps it is necessary to correctly focus the objectives of this revision. It should be noted that other revisions related to thermosensitive nanosystems have been found in the literature and it is suggested to add among the references, such as Sanchez-Moreno P. et al. (2018) doi: 10.3390/nano8110935, Shao P. et al. (2011) doi: 10.1155/2011/389640, Lopes C.M. et al. (2018) doi: 10.1016/B978-0-12-813689-8.00005-7. I would suggest to center and expand the description paragraphs of the thermosensitive nanosystems for moderate hyperthermia. The writing of the article needs revisions, for example "Cancer gástrico" in table 2. Line 143: Please, add DOX abbreviation for doxorubicin. Line159: The MRI abbreviation is being used without definition and introduced later in the manuscript (line 510). Line 182: MRI is an abbreviation for “Magnetic Resonance Imaging”, not “Magnetic Resonance”. Line 318: If the main objective is the one indicated at the end of paragraph 4 (see comment 1), please provide more details and references about "different drugs". Through 4.1.2: The paragraph is too short and doesn't provide exhaustive information about the core-shell nanoparticles. Please expand and include examples and references. Line 453: Please, add that the liposomes contain encapsulated doxorubicin. Through section 5: There are problems with the structure of the manuscript, which make it difficult to read. In sections 4.1.1 and 4.2, preclinical studies had already been introduced. Paragraph 5 again introduces preclinical studies. Please, reorganize section 5 (title and content). Line 519: The MR-HIFU abbreviation is being used without definition and introduced later in the manuscript (line 560). Line 522: Please, specify the abbreviation Gd(HPDO3A). Through section6: Please, specify why no clinical trials are described for the other thermosensitive systems described. Please review the abbreviations MR and MRI and unify the abbreviation type where possible. Finally, it is necessary to make a lot of corrections in the references. I only mention a few examples of the errors: some titles of the articles are missing (e.g. References 14, 27, etc.), some titles are incorrect (e.g. 34), there are striking repetitions in the lists of authors (e.g. 45, 78), in some articles the authors are badly cited (e.g. 42, 78, 79), the newspapers are incorrect (e.g. 47, 63), the DOI (e.g. 44) is missing or the indicated DOI is not correct (e.g. 27, 47, 54).
In summary, I would recommend this work after major revisions.
Reviewer 3 Report
This review paper deals with thermoresponsive or thermosensitive nanosystems for cancer treatment, particularly on hyperthermia. The topic is very interesting and relevant. I have few suggestions and comments to the authors:
In chapter 2 "Hyperthermia" please briefly mention the effects of hypothermia and other possible variations in tumor environment (e.g. hypoxia, pH, neovasculature, etc.) and their use in cancer treatment.
I am missing here further classes of thermosensitive polymers, exploiting the thermal sensitivity (~37°C) labelled with radionuclides for internal radionuclide therapy and radiolabelled magnetic NPs for multimodal treatment protocols.
Particularly the works of Hruby group focused on the radionuclide and chemo/radiotherapy delivery and immobilization based on thermosensitive polymers (at various LCST) should be mentioned. E.g.: New Binary Thermoresponsive Polymeric System for Local Chemoradiotherapy DOI:10.1002/app.29237, New bioerodable thermoresponsive polymers for possible radiotherapeutic applications DOI:10.1016/j.jconrel.2007.02.009, Thermoresponsive polymers as promising new materials for local radiotherapy DOI:10.1016/j.apradiso.2005.05.043, review DOI:10.1021/acs.langmuir.6b01527, etc.
Magnetic NPs labelled with radionuclides are also worth to include in this review within the magnetic field induced hyperthermia, since it may open novel multimodal treatment of tumors. Several papers appeared with various combinations of radionuclides, including alpha particle emitters.
E.g.:
Pospisilova et al.: 59Fe-labelled SPIONs, DOI:10.1007/s11051-016-3719-0
Wang et al.: 111In-labelled SPIONs, DOI:10.1016/j.nucmedbio.2014.08.014
Mokhodoeva et al.: 223Ra labelled SPIONs, DOI: 10.1007/s11051-016-3615-7
Ognjanovic et al.: 99mTc, 90Y, and 177Lu-labelled Iron Oxide NPs, DOI:10.1021/acsami.9b16428
Possible targeting of magnetic NPs with external magnetic field could be also mentioned (e.g. the work of De Simone et al.: DOI: 10.1002/cmmi.1718).
Please extend the chapter 7 "Conclusion". Please add an outlook to possible future developments in this field.
Spell check:
Lines 31-32: Please reformulate "nanosystem systems"
Lines 64/82/126/143/144/196: Please correct "37.5 °C" to "37.5°C"
Line 451: Please check the nomenclature.
Lines 462 / 588 : correct "thermos-sensitive"
Reviewer 4 Report
The review submitted by Pereira Gomes et al. to Pharmaceuticals describes various thermosensitive systems proposed for cancer treatment.
1) The great lack of this work is the almost total absence of references in the text. This makes it really difficult for a reviewer to judge the novelty and the accuracy of the work, even if references are reported at the end of the work.
2) The use of english should be accurately checked as there are many mistakes and non-sense sentences in the text that prevent fluid reading:
lines 228-230
lines 401-402
lines 449-450
lines 462-463: same sentence repeated in a few lines, review this paragraph.
Thermos-sensitive: this expression, used many times by the authors, does not exist. It should be thermo-sensitive, thermo-sensitive or thermo-responsive.
lines 530-532
and many others, please check the use of english.
Moreover, in order to improve the quality of the manuscript:
3) Table 1. Please add references and indicate the years in which clinical trials took place.
Line 185-187: please add reference
Line 195-196: add reference
4) describe with more details the HIFU technique (which intensity? why focused? how it works? is it applicable to all tissues and body regions?).
Line 218-223: add references
5) add a paragraph about thermosensitive hydrogels in part 4.
6) as polymeric micelles are assembled into this form only above CMC, please discuss their stability after in vivo injection.
7) Figure 2. define Tc (Critical Temperature): it could be not clear to non-expert readers, so it should be defined at least one time in the text or in the caption.
8) please briefly describe the mechanism of action of radio-frequency ablation. how is it obtained? How the setup was applied to patients?
Round 2
Reviewer 1 Report
The revised version of manuscript entitled “Thermosensitive nanosystems associated with hyperthermia for cancer treatments: a review” by Gomes et al. was improved based on the comments. However, further improvements are required based on following comments:
The title can be revised as “Thermosensitive nanosystems associated with hyperthermia for cancer treatments”, “: a review” is not required. References are lacking in many sentences. Also, references need to be added to first sentence of the paragraph if few sentences are added from same reference. Add references to the sentences ending in line 66, 70, 71, 76, 80 (page 2); line 128, 148 (page 4); line 249 (page 7); line 261, 271 (page 8); line 334 (page 9); line 355, 358, 368, 383, 385 (page 10); line 397, 416 (page 11); line 435, 452 (page 12); line 468, 473, 502 (page 13); line 517 (page 14); line 544, 553, 570 (line 15); line 583, 586, 596 (page 16); line 612, 613, 622 (page 17); line 650 (page 18); line 666, 673, 689 (page 19). Correct the sentence “….induce to direct cytotoxicity, as well as to lead to…” in line 85 (page 3) as “….induce direct cytotoxicity, as well as lead to….” In Figure 1, correct the spelling for “Apoptose” as “Apoptosis” and “Increased drug deliverance” as “Increased drug delivery”. In line 149 (page 4), revise the sentence “ ….was shown to have enhanced…” as “….was shown to induce enhanced…” In Table 1, what does “active” mean for response in reference 30, “any evident disease” mean for reference 31? In Table 1, revise the sentence “…do not increased toxicity…” as “….did not increase toxicity…” for response in reference 33. Correct the sentence “…can be also implemented…” in line 179 (page 6) as “….can also be implemented…” Correct the sentence “….to apply to heat…” in line 205 (page 6) as “…..to apply heat…” Provide full form for the abbreviation RFA in line 230 (page 7). Rephrase the sentence in line 307-308 (page 9) to make it clear. Correct the words “described” in line 327 and “reviewed” in line 328 as “describe” and “review”, respectively. Remove word “these” from the sentence in line 407 (page 11). Revise the sentence “….monomers have shown interesting as a temperature-sensitive material and that are attractive…” in line 412-413 (page 11) as “…monomers are attractive….” Add full form for the abbreviation “BGC” in line 425 (page 11). Correct “P-NIPA” in line 435 (page 12) as “P-NIPAm” In Table 2, correct “Docetaxel e Paclitaxel” as “Docetaxel and Paclitaxel”. In line 459 (page 12), provide some examples of other stimuli in brackets at the end of the sentence. Correct the sentence “…copolymer which DOX…” in line 476 (page 13) as “….copolymer to which DOX….” Replace the word “carrier” in line 502 (page 13) with “deliver”. Rearrange the words “liposome thermosensitive” in line 625 (Page 17) as “thermosensitive lipsome”. Remove the word “to” from the sentence in line 634 (page 17). Replace the word “doxorubicin” in line 648 (page 18) with abbreviation “DOX”. Revise the sentence “…of survival the combination…” in line 682 (page 19) as “….of survival with the combination…” Revise the sentence “….improvements in heating control.” In line 685 (page 19) as “….required improvements in heating control.”
Reviewer 2 Report
The authors have appropriately and exhaustively addressed the modifications therefore it is now ready for publication as is in Pharmaceuticals is recommended.
Author Response
Thanks for your comments.
Reviewer 4 Report
I have appreciated the revisions made to the manuscript, according to reviewers'suggestions. I think the quality of the manuscript has now been improved.
Author Response
Thanks for your comments.